# Association between Pet Ownership and Mental Health and Well-Being of Canadians Assessed in a Cross-Sectional Study during the COVID-19 Pandemic

**DOI:** 10.3390/ijerph19042215

**Published:** 2022-02-16

**Authors:** José Denis-Robichaud, Cécile Aenishaenslin, Lucie Richard, Marion Desmarchelier, Hélène Carabin

**Affiliations:** 1Independent Researcher, Amqui, QC G5J 2N5, Canada; josedr@hotmail.ca; 2Département de Pathologie et Microbiologie, Faculté de Médecine Vétérinaire, Université de Montréal, Saint-Hyacinthe, QC J2S 2M2, Canada; cecile.aenishaenslin@umontreal.ca; 3Centre de Recherche en Santé Publique (CReSP), Montreal, QC H3N 1X9, Canada; lucie.richard@umontreal.ca; 4Groupe de Recherche en Épidémiologie des Zoonoses et Santé Publique (GREZOSP), Saint-Hyacinthe, QC J2S 2M1, Canada; 5Faculté des Sciences Infirmières, Université de Montréal, Montreal, QC H3T 1J4, Canada; 6Département de Sciences Cliniques, Faculté de Médecine Vétérinaire, Université de Montréal, Saint-Hyacinthe, QC J2S 2M2, Canada; marion.desmarchelier@umontreal.ca; 7Département de Médecine Sociale et Préventive, École de Santé Publique, Université de Montréal, Montreal, QC H3N 1X9, Canada

**Keywords:** companion animal, generalized anxiety disorder, One Health, quality of life, stress

## Abstract

The objective of this cross-sectional study was to assess the association between pet ownership and quality of life (QoL), loneliness, anxiety, stress, overall health, and mental health of Canadians during the COVID-19 pandemic using a One Health perspective. An online bilingual survey was completed by 1500 Canadians in April–May 2021. Socio-demographics, health, QoL, stress and anxiety, loneliness, social support, pet ownership, and attitude towards pets data were collected. Crude and adjusted associations between pet ownership and mental health and well-being indicators were estimated. The 1500 participants were from all provinces and territories, half were women; half of the participants were pet owners by design. The crude association estimates showed that pet owners had poorer QoL, overall health, and mental health than non-pet owners, and were lonelier, more stressed, and more anxious than non-pet owners. Adjusted estimates showed that these associations disappeared with the inclusion of the confounders (socio-economic, demographic, health, and pet-related variables). Our results suggest that there was no association between pet ownership and the mental health and well-being indicators measured in the present study.

## 1. Introduction

The World Health Organization declared the 2019 novel coronavirus disease (COVID-19) a global pandemic in March 2020. Following a first wave primarily affecting the elderly population in Spring 2020 and a respite in transmission during summer 2020, many Canadian provinces and territories (re)-implemented strict public health measures starting in November 2020 to avoid overwhelming the health care system and reduce deaths as the second wave was growing [1]. The list and stringency of public health measures varied by province and region and included discouraging international travel, closure of non-essential businesses, self-isolation of those with symptoms suggestive of COVID-19 until testing negative. The measures also required citizens to self-distance, wear a face mask, and avoid indoor crowded space. Moreover, measures such as forbidding private indoor gatherings, implementing curfews, and encouraging all to work from and stay at home were put in place [1]. These measures have resulted, for some people, in increased social isolation, loneliness, and anxiety, which could have led to negative impacts on quality of life (QoL) and mental health [2,3].

Research has shown a lower proportion of Canadians reporting excellent or very good mental health during the COVID-19 pandemic when compared to the previous years [4]. Moreover, many Canadians reported their mental health to have worsened after the onset of physical distancing, and most (88%) reported at least one symptom of anxiety [5]. Several studies have assessed the association between risk factors and poorer well-being, QoL, and mental health during the pandemic in Canada and worldwide. Gender, visible minority groups, relationship status, socioeconomic status (SES), chronic illness, disabilities, and age were found to be associated with anxiety and poorer mental health [4,6,7,8,9,10].

Another factor possibly affecting mental health during the COVID-19 pandemic is pet ownership. A cross-sectional online survey conducted among a convenience sample of 1297 pet owners in Spain during the first wave showed that almost half of the respondents perceived their companion animals helped them more during the lockdown than before [11]. However, the authors also reported in their adjusted analyses (including all measured independent variables) that those perceiving that their companion animals supported them more through the confinement were less likely to have felt that their QoL had remained the same or improved since the pre-confinement period. In other words, those most affected by the confinement were also the ones who felt more supported by their pets during that time. The impact of pet or dog ownership on mental health has also been assessed since the beginning of the pandemic in Australia and the UK and in a multi-country (Europe, Americas, Asia, and Australia) study [12,13,14]. These studies found that companion animal owners perceived that their mental health improved due to their dog or the ability to walk them [12,13]. They also perceived that their mental health and loneliness had not worsened as much as that of non-pet owners [12]. However, the Australian study found that pet owners had worse QoL than non-pet owners and found no difference for resilience and loneliness [14]. These discrepancies could be due to different methodological factors or to differences in the populations studied. While the average self-reported mental health scores of the Canadian population has worsened since the beginning of the pandemic [4], it is unclear if the mental health of pet owners differed from that of non-pet owners during the COVID-19 pandemic period. 

A recent survey suggests that 38% and 41% of Canadian households have at least one cat and dog, respectively [15]. Many animal owners perceive that human–animal interactions have a positive impact on their physical and mental health [13,16,17]. However, the reported effect of pet ownership on the mental and physical health of elderly populations [18] or people living with mental health problems [16] has not always been positive. Moreover, dog ownership was not associated with lower levels of stress and depression in a longitudinal study conducted in the general population prior to the pandemic [19]. Qualitative work has found that the species of, the number of, the perceived friendliness of, and attachment to companion animals are factors that could influence the association between pet ownership and health (reviewed by Brooks et al., 2018 [16]). The positive impact of pet ownership seemed to be most important in times of crisis, but negative impacts can be caused by the financial, practical, and emotional burden of ownership or by the pain of losing a companion animal [20].

The objectives of this study were to assess the association between pet ownership and the QoL, overall health, loneliness, anxiety, stress, and mental health of Canadians during the confinement measures put in place in response to the second wave of COVID-19 in Canada.

## 2. Materials and Methods

### 2.1. Study Design, Study Population, and Sampling Strategy

A cross-sectional study was conducted using an online survey administered in collaboration with Leger Opinion (LEO; Montreal, QC, Canada) from 14 April to 5 May 2021. Potential participants were randomly selected from the firm panel (representative panel of the Canadian population), and participants received points (500 points, equivalent to approximately $0.50, per 5 min) that could be exchanged for cash, gift cards, or donation (https://www.legeropinion.com/fr/recompenses/, accessed on 7 December 2021). To be included, participants had to be adult (18 years and older) Canadian residents and had to complete the whole survey.

### 2.2. Questionnaire Development

The questionnaire was developed in English by a multidisciplinary team. The questions were selected to assess the outcomes of interest, as well as the potential confounders identified using directed acyclic graphs (DAG) created with DAGitty (available at dagitty.net/mCLVPSr, accessed on 13 February 2022) [21]. Validated questions and scales, as well as questions previously used by Statistics Canada, were used whenever possible to ensure the repeatability, comparability, and accuracy of the collected data. The final questionnaire (available at https://doi.org/10.7910/DVN/JKDKWY, accesses on 13 February 2022) included 80 close-ended questions and was translated into French. The questionnaire was available to participants in both French or English languages.

### 2.3. Measurement of the Outcomes of Interest: QoL, Overall Health, Loneliness, Anxiety, Stress, and Mental Health

Many validated tools have been used to measure QoL, loneliness, mental health, and well-being before and during the COVID-19 pandemic. The EQ-5D-5L, developed by the EuroQol group to assess QoL across different populations [22] including Canada [23], was used. This tool includes five questions regarding mobility, self-care, usual activities, pain and discomfort, and anxiety and depression, as well as one question on overall health. The EQ-5D-5L was used to assess QoL in China during the pandemic [8] but has not yet been used for this purpose in Canada. For the present study, the combined results across the five questions were translated on a standardized scale validated for Canadians (QoL utility score [23]). The participant’s self-rated overall health, measured in the EuroQol with a vertical visual analog scale from 0 to 100, was also assessed [22].

Loneliness was assessed using a 3-question scale [24]. The scale showed satisfactory reliability and validity among the elderly population. Each question can be answered with three options (hardly ever = 1, sometimes = 2, and often = 3), which, when summed over the three questions, is considered as the participant’s score (3 being the least lonely, and 9 being the loneliest).

Questions used by Statistics Canada during the COVID-19 pandemic to assess the perception of mental health and stress were used, in addition to the generalized anxiety disorder (GAD-7) score [25,26]. Here, the participants were asked to self-assess the amount of stress in their life and their mental health since the start of the COVID-19 pandemic in two distinct multiple-choice questions for stress (asking about the amount of stress in most days of their life, answering a 5-point scale from not at all stressful to extremely stressful) and mental health (asking to describe their mental health since the start of the COVID-19 pandemic, answering a 5-point scale from excellent to poor). The participants’ anxiety was assessed using the GAD-7 score. Briefly, the answers for each of the seven questions were scored as 0 (not at all), 1 (several days), 2 (more than half the days), and 3 (nearly every day). Scores of 5, 10, and 15 were used as thresholds for mild, moderate, and severe anxiety, respectively [25].

Together, these questions generated six scores which were analyzed as continuous outcomes (QoL (utility score; EQ-5D-5L), overall health (EQ-5D-5L), and loneliness [24]) or categorical outcomes (stress [26], mental health [26], and anxiety (GAD-7)).

### 2.4. Exposure and Potential Confounders and Effect Modifiers

The exposure of interest was pet ownership (reported presence of at least one companion animal in the participant’s household). While the companion animal species were recorded, analyses were conducted with pet ownership as a dichotomous variable.

Based on the initially developed DAGS, participants’ demographics, physical and mental health, current social support, attitude towards companion animals and recent changes in ownership, and employment status and change were considered as potential confounders for the relationship between pet ownership and the outcomes of interest.

The health questions were previously used during the pandemic to assess the impact of the COVID-19 pandemic on Canadians [26]. They included long-term conditions and disabilities, as well as physical and mental health perception since the beginning of the pandemic, and exposure to or illness from COVID-19 in the past 6 months.

The social support of the participants was assessed with three questions previously used as covariables to measure the impact of alcohol consumption on the mental state in a study conducted in Canada [27]. Participants were considered to have social support if they answered yes to the three questions or no social support if they answered yes to 0 to 2 questions. Finally, the attitude of people towards companion animals was quantified using the modified pet attitude scale (18-item Likert format self-report instrument [28,29]). The results were summed on an 18 to 126-points scale, 18 being the least and 126 being the most favorable attitude towards companion animals.

### 2.5. Sample Size and Statistical Analyses

A sample size of 1,500 participants was calculated to identify a 1-point difference (alpha = 0.01, power = 0.80) for the GAD-7 score [25] between pet owners and non-pet owners, as was found for physical activity levels in a recent Canadian study (variance = 25 [30]), accounting for covariables (adjustment factor = 1.33). As the participants were meant to be equally distributed between pet owners and non-pet owners, panel members were not eligible to participate when a total of 750 participants in either category was reached.

Statistical analyses were conducted using R (version 4.0.5) with the R Studio interface (version 1.3.1093 [31]). Descriptive statistics for each outcome, the exposure, and the potential confounders were first generated. Bayesian hierarchical multivariable ordinal and linear models were run for each outcome. For all Bayesian models, the brms package was used as the interface to the programming language Stan, which implements Hamiltonian Monte Carlo Sampler and the No-U-Turn Sampler [32]. Continuous scores (QoL utility, QoL overall health, and loneliness) were assessed using Bayesian hierarchical gaussian linear regression models [32], and categorized scores (stress, mental health, and anxiety) were assessed using Bayesian hierarchical cumulative ordinal models (probit link [33]). Provinces and territories were grouped in geographical regions as follows: British Columbia, Prairies and Territories (Alberta, Saskatchewan, Manitoba, Nunavut, Yukon, and Northwest Territories), Ontario, Québec, and Atlantic (New Brunswick, Nova Scotia, Prince Edward Island, and Newfoundland and Labrador). All models were run with the geographical region of the participants as a random-effect intercept and using weakly informative priors. Briefly, priors for *βs* were normal distributions centered on 0 with sigma = 1, while priors for SD or sigma were half Cauchy distributions centered on 0 with sigma = 2. The hyper distribution for the random-effect intercept was also a half Cauchy distribution centered on 0 with sigma = 2. Intercepts in linear regressions were normal distributions centered on the mean of the outcome variable (${\bar{x}}_{y}$) with sigma = $SD_{y}$, and intercepts in cumulative ordinal models were normal distributions centered on 0 with sigma = 1. Univariable models assessing the association between pet ownership and the six outcomes were first generated. Stratified models were built to assess if gender, disability, social support, and age modified the effect of pet ownership on the six outcomes, but no effect modification was identified.

The DAG initially developed to identify potential confounding variables to include in the questionnaire was updated by validating the conditional independence. The DAGitty interface [21] was used to identify the minimal sufficient adjustment sets for estimating the total effect of pet ownership on the six outcomes of interest.

Final multivariable models were built, including adjustment for the minimal sufficient set. The final models were generated using three chains with a length of 3000, in which the first 1000 iterations were used as warm-up [34]. Convergence was monitored through the visual inspection of trace plots of variance components, density plots, and by obtaining effective sample sizes (ESS). An ESS of 1000 or greater was considered sufficient for reaching convergence [32]. Linearity and homoscedasticity of the residuals were verified for the three linear models (QoL overall health and utility, and loneliness scores). For the three cumulative ordinal models (stress, mental health, and anxiety scores), the absence of category-specific effect and equality of variance were assessed, and models were adjusted to meet these assumptions [35]. Stratified models using only pet owners who reported having a dog, a cat, or both were also built to assess if pet owners of uncommon species influenced the results.

## 3. Results

The targeted 1500 participants were recruited, with an average completion time of 15 min (median of 11 min). A total of 1826 people started the survey, but 326 were not included as they did not consent to participate in the study (*n* = 134), were under 18 years old (*n* = 5), were not from Canada (*n* = 3), initiated the survey after the pet ownership quota had been reached (*n* = 115 pet owners), or did not complete the survey (*n* = 69). The participants who did not complete the survey or started after the pet ownership quota was reached (*n* = 184) were from all provinces and territories except Nunavut and Yukon, and from all age groups (did not differ from the participants included in the final dataset).

Table 1 describes the distribution of variables included in the DAG to identify the minimal sufficient sets for the six outcomes of interest among pet owners and non-pet owners. As expected, the demographic and socio-economic indicators were similar to that of the Canadian population. About half of the participants did not see a change in their household income since the start of the pandemic, but a decrease was reported by a little more than a quarter. Approximately 5% of respondents reported that at least one person had tested positive for COVID-19 in the past 6 months in their household. The participants’ attitude toward pets ranged from almost least favorable (score = 19) to most favorable (score = 126), with a higher median in pet owners (score = 107) than in non-pet owners (score = 86.5; difference of inverse-normal rank transformation = −0.85; 95% BCI= −0.94 to −0.76).

Based on the EQ-5D-5L, the participants’ QoL utility score ranged from −0.01 to 0.95 (median = 0.87; mean = 0.83) with a median self-assessed overall health score of 77 (mean = 73.5). The participants’ loneliness score ranged from 3 to 9 (median = 5, mean = 5.3). Participants perceived their mental health since the start of the COVID-19 pandemic as excellent (*n* = 157; 11%), very good (*n* = 406; 27%), good (*n* = 504; 34%), fair (*n* = 336; 22%), or poor (*n* = 97; 6%), and their life to be not at all (*n* = 173; 11%), not very (*n* = 474; 32%), a bit (*n* = 575; 38%), quite a bit (*n* = 236; 16%), or extremely (*n* = 42; 3%) stressful. The anxiety level of the participants, using the GAD-7, was categorized as minimal (*n* = 828; 55%), mild (*n* = 421; 28%), moderate (*n* = 147; 10%), or severe (*n* = 104; 7%). The distribution of these variables among pet owners and non-pet owners is described in Table 2.

The pet owners (*n* = 750) owned mainly dogs (*n* = 423) and cats (*n* = 407) but also birds (*n* = 27), fish (*n* = 24), rabbits (*n* = 19), rodents (*n* = 13), amphibians (*n* = 5), reptiles (*n* = 7), ferrets (*n* = 2), and horses (*n* = 1), and most of them (*n* = 704; 94%) had at least a cat or a dog. Most pet owners were the main caretakers of at least one of the companion animals in the household (*n* = 594; 79%). Most pet owners reported no difference in their relationship with their companion animals (*n* = 447; 60%) since before the COVID-19 pandemic while 37% reported that their relationship had improved. Thirteen percent (*n* = 100) of the pet owners acquired at least one of their animals after the beginning of the COVID-19 pandemic, and 12% (*n* = 173) of all the participants lost at least one animal in the year before the survey (approximately the beginning of the pandemic).

The univariable models showed that pet owners had poorer QoL utility scores, self-rated overall health and mental health, and greater loneliness, self-reported stress, and anxiety (Table 3 and Table 4). Descriptive statistics also suggested that pet owners were younger and that they had a lower level of education than non-pet owners (Table 1). Moreover, there were more women, Caucasian participants, people reporting no social support, and people with disabilities who owned a pet compared to those who did not. The multivariable models adjusted for the minimal sufficient adjustment sets resulted in posterior distributions not showing any strong associations between pet ownership and the different outcomes, suggesting that the observed univariable associations were due to confounding (spurious associations; Table 3 and Table 4). For example, in the unadjusted model for loneliness, pet owners had on average a score of 0.31 (95% BCI = 0.11; 0.51) higher (lonelier) than non-pet owners (Table 3). However, there was no difference between pet owners and non-pet owners in the adjusted model (estimate = −0.10; 95% BCI = −0.30; 0.09). Similar results were found for all outcomes. For mental health, the prevalence odds of being in one worse category level of mental health increased by 1.25 (95% BCI: 1.12; 1.38) for pet owners compared to non-pet owners in the crude model. Such association disappeared in the multivariable model (posterior OR = 0.97; 95% BCI = 0.85; 1.11). Models including cat and dog owners separately found similar estimates (same patterns of associations in the crude and adjusted models; summarized in Appendix A) to the full models.

## 4. Discussion

Our study demonstrated that, when the analyses are appropriately adjusted for confounding, pet owners did not report worse indicators of mental health and well-being than non-pet owners during the second wave of the COVID-19 pandemic in Canada. Moreover, our results clearly show that a set of confounding factors are involved here that, if ignored, would have given the impression that pet owners have worse outcome than non-pet owners. This is because a greater proportion of pet owners were more often females, less educated, without a social network, and with disabilities, all characteristics documented as risk factors for anxiety and poorer mental health [4,6,7,8,9,10].

Studies measuring the association between pet ownership and mental health or well-being status during the pandemic have found conflicting results. Two studies reported that pet owners perceived that their mental tensions diminished due to the presence of their dog and the ability to walk them in addition to reporting a smaller decrease in mental health and a smaller increase in loneliness during lockdown [12,13]. In contrast, another study found pet ownership to be associated with poorer QoL but not with resilience and loneliness [14]. Multiple reasons could have led to these differences, including different studied populations and contexts (Australia and UK), use of different mental health and well-being scales, and the inclusion of different confounder sets in the final models [12,14].

A longitudinal study conducted prior to the pandemic found no difference between dog owners and non-dog owners for stress and depression [19]. In other studies, the positive impact of pet ownership was limited to specific situations such as managing mental health conditions [36,37,38]. Considering the COVID-19 pandemic a time of crisis, we expected that pet owners would, in general, cope better with the situation and have better QoL and mental health than non-pet owners. Multiple mechanisms have been suggested to explain the positive impact of pet ownership on mental health (reviewed by McCune et al., 2014 [39]). Indeed, companion animals are providing social support that can act as a buffer against stressors. Moreover, it has been suggested that human–animal interactions are associated with hormonal changes such as the release of oxytocin and a decrease in cortisol when humans interact with their companion animals. Having a dog also leads to increased physical activity through dog walking and leads to positive human interactions. However, the cross-sectional nature of our study meant that it was impossible to determine if pet ownership dampened the impact of the pandemic on mental health and QoL or not. A longitudinal study assessing the changes throughout an ordeal such as a pandemic or assessing changes following the adoption of a pet would have allowed for a better understanding of the underlying patterns.

Our findings showed that there was no difference for mental health and QoL when pet owners and non-pet owners were compared. While the mechanisms underlying these results were not explored in the present study, previous studies showed there are concerns associated with pet ownership, including financial, logistical, and emotional stressors [40,41]. Given the pandemic context at play here, it is possible that such concerns and worries could have negatively influenced mental health issues and offset or attenuate the positive effects of pet ownership.

The inclusion of the minimal adjustment sets was likely important in our findings as their association with mental health and well-being status have been identified in multiple contexts, including since the beginning of the pandemic [42,43,44]. This highlights the complexity of the human–animal relationships and their impact on mental health. Future research should aim at unraveling the complex patterns underlying this relationship and include variables likely to act as confounders, intermediaries, or effect modifiers.

The target population was the Canadian population during the pandemic in this study. The panel used to recruit participants resulted in a representative sample of the Canadian population, with a proportion of gender, age, household income, and ethnicity similar to the 2016 national census [45], even though the sampling was such that the same number of pet owners and non-pet owners were included. There was, however, a greater proportion of people with a University degree and a smaller proportion of people without a high school diploma or certificate in our sample than in the 2016 census [45]. It is unclear how this discrepancy could have influenced our findings, but our conclusion might not be generalizable to the whole Canadian population. Another indicator that the sample of participants in the present study might differ from the Canadian population is the results from a questionnaire on mental health conducted by Statistics Canada during the beginning of the pandemic [5]. While a similar proportion of participants to both surveys perceived they had a fair or poor mental health (present study: 29%, Statistics Canada: 24%), a smaller proportion of the participants of the present study reported having a high level of stress (present study: 19%, Statistics Canada: 28%) or had symptoms of moderate or severe anxiety (GAD-7; present study: 17%, Statistics Canada: 41%).

Several pet owners tried to enter the survey after the quota of participants for that category had been reached, which was not the case for non-pet owners. This could have happened because there are more pet owners in the Canadian population than non-pet owners [15], which could have resulted in a selection bias. Our results could have been affected by selection bias if pet owners with poorer mental health and quality of life were more prone to participate in the survey. Though possible, the impact of this bias is likely to be limited since the invitation to participate in the survey did not mention pet ownership but did mention mental health and QoL. In addition, through our thorough identification of potential confounders, factors that could have pushed a larger proportion of pet owners with poorer mental health or QoL to participate in the study would have been measured and adjusted for in the analyses [5,15].

Our results could have also been affected by misclassification and measurement error since mental health, QoL, and well-being are difficult to measure. Some of the tools included in the present study have been validated (EQ-5D-5L, loneliness scale, and GAD-7 [22,23,24,25]), but other questions were not (level of stress and perceived mental health). While we collected valuable information with these tools and questions, we could have missed subtleties of mental health and well-being that could be assessed with qualitative work. Moreover, the EQ-5D-5L utility score has been shown to be influenced by the heterogeneity in respondents’ ordinal preferences, which suggested that small differences (less than 0.05) are negligible to support a difference in QoL [46]. The impact that information bias could have had on our results is difficult to predict and may have been differential. Indeed, perhaps pet owners are more prone to admit to mental distress or more aware of limitations in their quality of life, which could have hidden a positive association between pet ownership and mental health or quality of life.

Our findings could have also been influenced by the human–animal bond, as the strength of this bond was associated with poorer mental health [11,12]. This was not directly assessed in the present study, but the pet attitude score was included as a confounding variable in all minimal adjustment sets and is likely a proxy for the human–animal bond. The pet attitude scale is a tool that assesses favorableness of attitudes toward pets [29], and even though it has not been evaluated, people with stronger human–animal bonds may also be more favorable towards companion animals.

Finally, none of the included covariables were identified as effect modifiers, and the results remained the same when analyses were conducted on separately for cat and dog owners. This suggests the results are similar across the sample and no subgroup benefited more or less than the whole sample. Previous studies showed there was a positive impact of pet ownership on the mental and physical health of the elderly population and the population living with mental health problems [16,18]. It is unclear why these associations were not found in the present study as the positive impact of pet ownership seemed to be most important in times of crisis [16].

## 5. Conclusions

Overall, pet ownership was not associated with mental health and well-being indicators during the COVID-19 pandemic. There were, however, multiple factors associated with both pet ownership status and mental health outcomes, resulting in spurious crude associations suggesting pet owners had decreased QoL, mental health, and overall health and increased level of stress, anxiety, and loneliness. It remains unclear if pet owners benefited from the presence of their companion animals during the pandemic which could be assessed via longitudinal and qualitative designs.

## Figures and Tables

**Table 1 ijerph-19-02215-t001:** Distribution of the variables included in the directed acyclic graph (DAG) to identify the minimal sufficient adjustment set for estimating the total effect of pet ownership on the different health and well-being outcomes assessed in a cross-sectional study on 1500 Canadians during the COVID-19 pandemic (14 April to 5 May 2021).

Variables	% (n) or Median (Range)	% or Median (95% CI ^1^)	
	All	Pet Owners	Non-Pet Owners
Province and territories †			
British Columbia	12.5% (187)	10.4% (8.2; 12.6)	14.6% (12.0; 17.1)
Alberta	9.9% (149)	10.0% (7.9; 12.1)	9.9% (7.7; 12.0)
Saskatchewan	3.3% (49)	3.2% (1.9; 4.5)	3.3% (2.0; 4.6)
Manitoba	3.3% (49)	4.0% (2.6; 5.4)	2.5% (1.4; 3.7)
Ontario	38.3% (575)	37.2% (33.7; 40.7)	39.5% (36.0; 43.0)
Québec	25.8% (387)	27.2% (24.0; 30.4)	24.4% (21.3; 27.5)
New Brunswick	2.4% (36)	3.2% (1.9; 4.5)	1.6% (0.7; 2.5)
Nova Scotia	2.5% (37)	2.4% (1.3; 3.5)	2.5% (1.4; 3.7)
Prince Edward Island	0.2% (3)	0.4% (0.0; 0.9)	0
Newfoundland and Labrador	1.2% (18)	1.2% (0.4; 2.0)	1.2% (0.4; 2.0)
Nunavut	0.1% (1)	0	0.1% (0.0; 0.4)
Yukon	0.4% (6)	0.5% (0.0; 1.1)	0.3% (0.0; 0.6)
Northwest Territories	0.2% (3)	0.3% (0.0; 0.6)	0.1% (0.0; 0.4)
Age *			
18 to 24 years old	8.5% (127)	9.6% (7.5; 11.7)	7.3% (5.5; 9.2)
25 to 34 years old	15.3% (229)	17.5% (14.7; 20.2)	13.1% (10.7; 15.5)
35 to 44 years old	18.5% (277)	20.2% (17.4; 23.1)	16.7% (14.0; 19.3)
45 to 54 years old	20.5% (308)	23.9% (20.8; 26.9)	17.2% (14.5; 19.9)
55 to 64 years old	17.3% (259)	16.1% (13.5; 18.8)	18.4% (15.6; 21.2)
65 years old and older	20.0% (300)	12.7% (10.3; 15.0)	27.3% (24.1; 30.5)
Gender *			
Women	50.2% (752)	53.6% (50.0; 57.2)	46.7% (43.1; 50.2)
Men	49.5% (743)	45.9% (42.3; 49.4)	53.2% (49.6; 56.8)
Other or did not answer	0.3% (5)	0.5% (0.0; 1.1)	0.1% (0.0; 0.4)
Highest level of education *			
Before high school	2.5% (38)	2.9% (1.7; 4.1)	2.1% (1.1; 3.2)
High school	21.1% (317)	22.9% (19.9; 25.9)	19.3% (16.5; 22.2)
College	32.4% (486)	36.4% (33.0; 39.8)	28.4% (25.2; 31.6)
University	43.9% (659)	37.8% (34.3; 41.2)	50.2% (46.6; 53.7)
Ethnicity *			
Caucasian	77.9% (1168)	83.3% (80.7; 86.0)	72.4% (69.2; 75.6)
Others ^2^	22.1% (332)	16.7% (14.0; 19.3)	27.6% (24.4; 30.8)
Annual household income *			
$19,999 or less	4.9% (74)	5.2% (3.6; 6.8)	4.7% (3.2; 6.2)
$20,000–$39,999	11.7% (176)	13.1% (10.7; 15.5)	10.4% (8.2; 12.6)
$40,000–$59,000	15.2% (228)	14.1% (11.6; 16.6)	16.3% (13.6; 18.9)
$60,000–$79,999	13.6% (204)	13.6% (11.1; 16.1)	13.6% (11.1; 16.1)
$80,000–$99,999	14.1% (211)	12.1% (9.8; 14.5)	16.0% (13.4; 18.6)
$100,000–$119,999	11.3% (170)	12.5% (10.2; 14.9)	10.1% (8.0; 12.3)
$120,000 or more	18.5% (278)	19.4% (16.5; 22.2)	17.7% (15.0; 20.5)
Prefer not to answer	10.6% (159)	10.0% (7.9; 12.1)	11.2% (8.9; 13.5)
Income change since the pandemic			
Did not change	50.9% (763)	48.0% (44.4; 51.6)	53.8% (50.2; 57.3)
Decreased	28.4% (426)	30.3% (27.0; 33.6)	26.5% (23.4; 29.7)
Increased	14.7% (220)	15.5% (12.9; 18.1)	13.9% (11.4; 16.3)
Did not know	2.3% (35)	2.5% (1.4; 3.7)	2.1% (1.1; 3.2)
Did not answer	3.7% (56)	3.7% (2.4; 5.1)	3.7% (2.4; 5.1)
Had a social support *			
Yes	75.7% (1135)	73.9% (70.7; 77.0)	77.5% (74.5; 80.5)
No	24.3% (365)	26.1% (23.0; 29.3)	22.5% (19.5; 25.5)
Tested positive to COVID-19 in the last 6 months (or someone in their household)			
No	95.2% (1428)	93.9% (92.1; 95.6)	96.5% (95.2; 97.8)
Yes	4.8% (72)	6.1% (4.4.; 7.9)	3.5% (2.2; 4.8)
Identified as a person with a disability			
No	87.5% (1312)	84.0% (81.4; 86.6)	90.9% (88.9; 93.0)
Yes	12.5% (188)	16.0% (13.4; 18.6)	9.1% (7.0; 11.1)
Had an emotional, psychological, or mental health conditions			
No	77.4% (1161)	71.5% (68.2; 74.7)	83.3% (80.7; 86.0)
Yes	22.6% (339)	28.5% (25.3; 31.8)	16.7% (14.0; 19.3)
Mental health change (compared to before the pandemic) *			
Much better	1.9% (28)	2.0% (1.0; 3.0)	1.7% (0.8; 2.7)
Somewhat better	6.9% (104)	7.6% (5.7; 9.5)	6.3% (4.5; 8.0)
About the same	47.6% (714)	41.9% (38.3; 45.4)	53.4% (49.8; 56.9)
Somewhat worse	36.7% (551)	40.1% (36.6; 43.6)	33.3% (30.0; 36.7)
Much worse	6.9% (103)	8.4% (6.4; 10.4)	5.3% (3.7; 6.9)
Pet change *			
No change	84.0% (1259)	75.1% (72.1; 78.3)	92.7% (90.8; 94.5)
Lost a pet in the last year	9.4% (141)	11.5% (9.2; 13.7)	7.3% (5.5; 9.2)
Acquired a pet since the start of the pandemic	4.5% (68)	9.1% (7.0; 11.1)	-
Lost and acquired a pet in the last year	2.1% (32)	4.3% (2.8; 5.7)	-
Number of people in the household *	2 (1 to 14)	2 (2; 3)	2 (2; 2)
Pet attitude *	97.5 (19 to 126)	107 (106; 109)	86.5 (85; 89)

† Considered as a grouping in multilevel models by geographical regions: British Columbia, Prairies, and Territories (Alberta, Saskatchewan, Manitoba, Nunavut, Yukon, and Northwest Territories), Ontario, Québec, and Atlantic (New Brunswick, Nova Scotia, Prince Edward Island, and Newfoundland and Labrador). * Included in the minimal sufficient adjustment set for estimating the total effect of pet ownership on the different outcomes. ^1^ 95% confidence intervals. ^2^ Included First Nations, Métis or Inuk, South Asian, Chinese, Black, Filipino, Arab, Latin American, Southeast Asian, West Asian, Korean, Japanese, and others.

**Table 2 ijerph-19-02215-t002:** Distribution of the outcome variables (health and well-being), assessed in a cross-sectional study on 1500 Canadians during the COVID-19 pandemic (14 April to 5 May 2021).

Variables	% (n) or Median (Range)	% or Median (95% CI ^1^)	
	Total	Pet Owners	Non-Pet Owners
Quality of life	0.87 (−0.01 to 0.95)	0.87 (0.87; 0.87)	0.90 (0.87; 0.91)
Self-assessed overall health	77 (0 to 100)	75 (73; 77)	79 (78; 80)
Loneliness	5 (3 to 9)	5 (5; 6)	5 (5; 5)
Perceived mental health			
Excellent	10.5% (157)	10.0% (7.9; 12.1)	10.9% (8.7; 13.2)
Very good	27.1% (406)	22.5% (19.5; 25.5)	31.6% (28.3; 34.9)
Good	33.5% (504)	34.0% (30.6; 37.4)	33.2% (29.8; 36.6)
Fair	22.4% (336)	25.9% (22.7; 29.0)	18.9% (16.1; 21.7)
Poor	6.5% (97)	7.6% (5.7; 9.5)	5.3% (3.7; 6.9)
Self-reported level of stress			
Not at all	11.5% (173)	9.3% (7.3; 11.4)	13.7% (11.3; 16.2)
Not very	31.6% (474)	28.3% (25.0; 31.5)	34.9% (31.5; 38.3)
A bit	38.4% (575)	38.0% (34.5; 41.5)	38.8% (35.2; 42.2)
Quite a bit	15.7% (236)	20.1% (17.3; 23.0)	11.3% (9.1; 13.6)
Extremely	2.8% (42)	4.3% (2.8; 5.7)	1.3% (0.5; 2.2)
Anxiety			
Minimal	55.2% (828)	48.3% (44.7; 51.8)	62.1% (58.7; 65.6)
Mild	28.1% (421)	30.9% (27.6; 34.2)	25.2% (22.1; 28.3)
Moderate	9.8% (147)	11.7% (9.4; 14.0)	7.9% (5.9; 9.8)
Severe	6.9% (104)	9.1% (7.0; 11.1)	4.8% (3.3; 6.3)

^1^ 95% confidence intervals.

**Table 3 ijerph-19-02215-t003:** Crude and adjusted^1^ estimated median (95% Bayesian Credible Intervals; BCI) of the effect of pet ownership * on quality of life utility score, overall health score, and loneliness score using cross-sectional data from a survey of 1500 Canadian (14 April to 5 May 2021).

	Univariable Models			Multivariable Models ^1^		
	Estimate	95% BCI	ESS ^2^	Estimate	95% BCI	ESS ^2^
Quality of life	−0.03	−0.05; −0.02	3679	−0.01	−0.02; 0.01	4626
Overall health	−2.0	−3.2; −0.7	4098	−1.2	−2.5; 0.1	4723
Loneliness	0.31	0.11; 0.51	4496	−0.10	−0.30; 0.09	5060

* Reference: non-pet owners. ^1^ Bayesian gaussian linear regression models including geographical region as a random-effect intercept. The multivariable model included the minimal sufficient adjustment set for estimating the total effect of pet ownership (age, gender, highest level of education, ethnicity, annual household income, social support, disability, current mental health change, pet change in the previous year, number of people in the household, and pet attitude score). ^2^ Estimation of the effective sample size (number of independent samples from the posterior distribution that would be expected to yield the same standard error of the posterior mean).

**Table 4 ijerph-19-02215-t004:** Crude and adjusted ^1^ posterior odds ratio (OR) and 95% Bayesian credible intervals (BCI) of the effect of pet ownership * on perceived mental health, self-reported stress, and anxiety using cross-sectional data from a survey of 1500 Canadian (14 April to 5 May 2021).

	Univariable Models			Multivariable Models ^1^		
	OR	95% BCI	ESS ^2^	OR	95% BCI	ESS ^2^
Perceived mental health	1.25	1.12; 1.38	4133	0.97	0.85; 1.11	4611
Self-reported stress	1.40	1.26; 1.55	4458	1.08	0.96; 1.23	4554
Anxiety	1.42	1.26; 1.60	4311	1.12	0.96; 0.30	4708

* Reference: non-pet owners. ^1^ Bayesian gaussian linear regression models including geographical region as a random-effect intercept. The multivariable model included the minimal sufficient adjustment set for estimating the total effect of pet ownership (age, gender, highest level of education, ethnicity, annual household income, social support, disability, current mental health change, pet change in the previous year, number of people in the household, and pet attitude score). ^2^ Estimation of the effective sample size (number of independent samples from the posterior distribution that would be expected to yield the same standard error of the posterior mean).

## Data Availability

The data and R code used to produce this manuscript are available in the Appendix A (Dataverse).

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
