# Peer review of "Association between Pet Ownership and Mental Health and Well-Being of Canadians Assessed in a Cross-Sectional Study during the COVID-19 Pandemic"

_ijerph, 2022, doi:10.3390/ijerph19042215_

Round 1

Reviewer 1 Report

The survey was designed well and conducted well.  As the authors mention in the conclusion that pet ownership was not associated with mental health and well-being during the COVID-19 pandemic.  However, they infer that pet ownership is the key factor for the low quality of life.  More solid explanations should be provided to reach that conclusion.  The level of education or age of participants or some other factors may be related to the quality of life rather than the pet ownership.  With further explanations, it would be of interest to the public that owning pet may not solve mental health issues or high quality of life.  Any suggestions for future studies to improve the research design?

Author Response

The survey was designed well and conducted well.  As the authors mention in the conclusion that pet ownership was not associated with mental health and well-being during the COVID-19 pandemic.  However, they infer that pet ownership is the key factor for the low quality of life.  More solid explanations should be provided to reach that conclusion.  The level of education or age of participants or some other factors may be related to the quality of life rather than the pet ownership.  With further explanations, it would be of interest to the public that owning pet may not solve mental health issues or high quality of life.  Any suggestions for future studies to improve the research design?

AU: Dear reviewer, thank you for taking the time to review this manuscript.
It is unfortunate our conclusions suggested we inferred that pet ownership was a key factor of low quality of life. We rephrased multiple sentences to make sure the emphasis was on the  absence of association between pet ownership and the six measured outcomes (spurious crude association; L29-32, L256, L286-287).

For the characteristics of pet owners (e.g., level of education and age), they were associated with both quality of life (and other outcomes) and pet ownership, as suggested by the confidence in Table 1.

We think a better understanding of the impact of pet ownership on mental health and well-being would be best assessed with longitudinal studies (L296-297, L376). This would also help identify if pet ownership contribute or not to better mental health or quality of life. While it is unlikely pet ownership can solve mental health issues, the cross-sectional nature of the present study does not allow us to comment on this, even with better explanations.

Reviewer 2 Report

The current study aimed at investigating whether there was an association between pet ownership and the well-being of the Canadian population during the COVID-19 pandemic. To that end, an online survey was applied to 1500 Canadians.

The study was carefully designed and conducted. However, I have some concerns regarding data analysis and the conclusions drawn that need to be addressed.

Firstly, at least as it is currently presented, I do not understand the value of the crude analysis that is reported. Given that the two groups (pet owners and non-pet owners) differed in characteristics (gender, age, disabilities) that are already known to affect some of the outcomes measured (anxiety, mental health), I cannot see the added benefit of analyzing the data (and presenting the results) of a crude analysis where these factors are not considered as confounders – since they will not reveal any “true” association between pet ownership and well-being. When confounders were included in the analysis, results showed no association between pet ownership and well-being outcomes. Together these results strongly point to the possibility that the worse well-being outcomes observed in pet owners were most likely due to other differences between the groups (gender, age, disabilities, and social support) than to pet ownership itself.

This same concern applies to data interpretation. Examples:

  • “which would increase the observed association between being a pet owner and having worse mental health and quality of life” (lines 340-341) - This relationship between owning a pet and having worse outcomes may not be “real” because of the confounders.
  • “associations suggesting pet owners had decreased QoL, mental health, and overall health and increased level of stress, anxiety, and loneliness” (lines 364-367) – These factors are not necessarily associated with pet ownership. Because the groups were unbalanced, it may be the case that gender, social support and disabilities were the factors associated with decreased well-being and that pet ownership did not play any role.

The final sentence of the abstract “Our results suggest that that people who adopt a companion animal have characteristics generally associated with poorer mental health” is also overstating. Besides the aforementioned problem, in this case authors are also commenting on causality, which is something that cannot be derived from the current study.

Secondly, I am also concerned about including cats and dogs together. Is there evidence that supports that cats and dogs have usually the same effect on owners’ well-being? Because I would expect them to be different. Related to this, in lines 352-353, it is not clear to me what the authors mean. What does “on only cat and dog owners” mean? Were analysis conducted in cat owners and dog owners separately?

Minor issues:

  • Inconsistency in reporting the outcomes: sometimes the authors report five (lines 74-76 and line 95), sometimes six (lines 120-122).
  • Line 137: the reference should be reported with a number.
  • Lines 342-343: “Our findings could also be influenced by the strength of the human-animal bond as it was associated with poorer mental health [11,32]” – This sentence is not clear. What does “it” refer to?

Author Response

The current study aimed at investigating whether there was an association between pet ownership and the well-being of the Canadian population during the COVID-19 pandemic. To that end, an online survey was applied to 1500 Canadians.

The study was carefully designed and conducted. However, I have some concerns regarding data analysis and the conclusions drawn that need to be addressed.

AU: Dear reviewer,
Thank you for taking the time to review this manuscript. We considered your comments and suggestions carefully and tried to integrate them, or to better justify our decisions, to this new version.

Firstly, at least as it is currently presented, I do not understand the value of the crude analysis that is reported. Given that the two groups (pet owners and non-pet owners) differed in characteristics (gender, age, disabilities) that are already known to affect some of the outcomes measured (anxiety, mental health), I cannot see the added benefit of analyzing the data (and presenting the results) of a crude analysis where these factors are not considered as confounders – since they will not reveal any “true” association between pet ownership and well-being. When confounders were included in the analysis, results showed no association between pet ownership and well-being outcomes. Together these results strongly point to the possibility that the worse well-being outcomes observed in pet owners were most likely due to other differences between the groups (gender, age, disabilities, and social support) than to pet ownership itself.

AU: We agree that our findings show there was no association between pet ownership and the measured outcomes. We changed the phrasing throughout the manuscript (L29-32, L256, L286-287) to better represent this, but kept the crude associations as suggested by the STROBE reporting guidelines (https://doi.org/10.2471/BLT.07.045120).

This same concern applies to data interpretation. Examples:

    “which would increase the observed association between being a pet owner and having worse mental health and quality of life” (lines 340-341) - This relationship between owning a pet and having worse outcomes may not be “real” because of the confounders.

AU: This is true, we adjusted the potential impact of this bias accordingly with our main conclusion (L342-344)

    “associations suggesting pet owners had decreased QoL, mental health, and overall health and increased level of stress, anxiety, and loneliness” (lines 364-367) – These factors are not necessarily associated with pet ownership. Because the groups were unbalanced, it may be the case that gender, social support and disabilities were the factors associated with decreased well-being and that pet ownership did not play any role.

AU: We agree that the association between pet ownership and these outcomes are due to the confounders, hence the sentence you are referring to saying the crude associations are spurious “resulting in spurious crude associations suggesting respondents who owned pet also had decreased QoL, mental health, and overall health and increased level of stress, anxiety, and loneliness”.

The final sentence of the abstract “Our results suggest that that people who adopt a companion animal have characteristics generally associated with poorer mental health” is also overstating. Besides the aforementioned problem, in this case authors are also commenting on causality, which is something that cannot be derived from the current study.

AU: We rephrased to address this (L29-32)

Secondly, I am also concerned about including cats and dogs together. Is there evidence that supports that cats and dogs have usually the same effect on owners’ well-being? Because I would expect them to be different. Related to this, in lines 352-353, it is not clear to me what the authors mean. What does “on only cat and dog owners” mean? Were analysis conducted in cat owners and dog owners separately?

AU: Indeed, we built stratified models for cat owners, dog owners, and cat and dog owners, but we obtained the same results. This was not detailed as we are not presenting the results, and we now corrected it (L190, L263, L355)

Minor issues:

    Inconsistency in reporting the outcomes: sometimes the authors report five (lines 74-76 and line 95), sometimes six (lines 120-122).

AU: the overall health was added to represent the 6 outcomes (L26, L75, L95)

    Line 137: the reference should be reported with a number.

AU: it seems some references were not well converted. This was changed (L123, L139 and following references)

    Lines 342-343: “Our findings could also be influenced by the strength of the human-animal bond as it was associated with poorer mental health [11,32]” – This sentence is not clear. What does “it” refer to?

AU: The sentence was rephrased for clarity (L349-350)

Round 2

Reviewer 1 Report

I have checked the changes made by the authors.  As the result, more discussion would be need to make this paper if interest to the readers.

Author Response

Dear reviewer,

Thank you for taking the time to review the manuscript. We believe the current version presents the research and results clearly, and discusses strengths and limitations of the study. It is not clear to us what additional discussion is required to make it of interest. We will be happy to make modifications to specific topics or comments.

Best regards,